# Virtual Screening and Molecular Dynamics Simulation Study of Influenza Polymerase PB2 Inhibitors

**DOI:** 10.3390/molecules26226944

**Published:** 2021-11-17

**Authors:** Keli Zong, Lei Xu, Yuxin Hou, Qian Zhang, Jinjing Che, Lei Zhao, Xingzhou Li

**Affiliations:** 1Chemical Engineering and Environmental Engineering, College of Chemistry, Liaoning Shihua University, Fushun 113001, China; zongkeliya@sina.com; 2Beijing Institute of Pharmacology and Toxicology, Beijing 100850, China; 15155969871@163.com; 3Tianjin Children Hospital, Tianjin Medical University, Tianjing 300074, China; yuxinhou968@163.com; 4Department of Medicinal Chemistry, School of Pharmacy, Fudan University, Shanghai 201203, China

**Keywords:** virtual screening, influenza virus, polymerase basic protein 2, inhibitor, molecular dynamics simulations

## Abstract

Influenza A virus is the main cause of worldwide epidemics and annual influenza outbreaks in humans. In this study, a virtual screen was performed to identify compounds that interact with the PB2 cap-binding domain (CBD) of influenza A polymerase. A virtual screening workflow based on Glide docking was used to screen an internal database containing 8417 molecules, and then the output compounds were selected based on solubility, absorbance, and structural fingerprints. Of the 16 compounds selected for biological evaluation, six compounds were identified that rescued cells from H1N1 virus-mediated death at non-cytotoxic concentrations, with EC50 values ranging from 2.5–55.43 μM, and that could bind to the PB2 CBD of H1N1, with Kd values ranging from 0.081–1.53 μM. Molecular dynamics (MD) simulations of the docking complexes of our active compounds revealed that each compound had its own binding characteristics that differed from those of VX-787. Our active compounds have novel structures and unique binding modes with PB2 proteins, and are suitable to serve as lead compounds for the development of PB2 inhibitors. An analysis of the MD simulation also helped us to identify the dominant amino acid residues that play a key role in binding the ligand to PB2, suggesting that we should focus on increasing and enhancing the interaction between inhibitors and these major amino acids during lead compound optimization to obtain more active PB2 inhibitors.

## 1. Introduction

Influenza is an acute respiratory infectious disease caused by the influenza virus. Influenza virus is a representative species of the Orthomyxoviridae family, which includes human and animal influenza viruses [1]. Influenza is classified into four types: Flu A–D, based on the different antigenic determinants on the nucleoprotein (NP) and matrix protein 1 (M1) of the virus, of which only Flu A and Flu B are considered important for human health. Flu A and Flu B infections in seasonal epidemics lead to 3–5 million cases of severe illness, and 250,000–500,000 deaths annually worldwide. Flu A is also responsible for occasional pandemics that usually involve contagious respiratory illness, with high morbidity and mortality rates [2]. To date, only a few anti-influenza drugs have been licensed to treat influenza infections, including neuraminidase (NA) inhibitors (oseltamivir, zanamivir, peramivir), M2 ion channel inhibitors (amantadine, rimantadine) and polymerase inhibitors (favipiravir, baloxavir) [3], although all have limitations. In general, their therapeutic effects or clinical safety need to be improved. The NA inhibitors have only a moderate impact on the severity of influenza symptoms and the duration of illness, and these drugs must be administered within 24–48 h of infection to achieve noticeable results [4]. Baloxavir has a greater effect on reducing virus replication and shedding than oseltamivir, but its effect on shortening the duration of symptoms is similar to oseltamivir [5]. Favipiravir has been approved as an antiviral drug stockpile in Japan; however, its use is limited due to teratogenicity [3]. In addition, anti-influenza drugs are limited by the rise in drug resistance. The use of M2 ion channel inhibitors has been limited due to the emergence and spread of resistance amongst circulating strains [6]. Strains resistant to the neuraminidase inhibitor oseltamivir phosphate have been widely reported [7]. Similarly, a drug-resistant strain of baloxavir appeared soon after its approval [8]. Consequently, it is necessary to develop new anti-influenza drugs with novel mechanisms of action, good therapeutic effects, and low levels of drug resistance.

The influenza viral polymerase has emerged as a promising target for influenza infection treatment. It comprises three subunits, PB1, PB2 and PA, that are responsible for the replication and transcription of the eight separate segments of the viral RNA genome in the nuclei of infected host cells [9]. Influenza does not possess cap-synthesizing machinery, and instead has developed a virus-mediated cap-snatching mechanism, stealing its cap from host mRNA during the transcription process [10]. Of the three domains located in the influenza virus PB2 subunit, the PB2 cap-binding domain can initiate ‘cap-snatching’, and then direct the capped primer; first toward the endonuclease site, and then into the polymerase active site for virus transcription [11]. Small molecules completely bound to the cap-binding domain inhibit mRNA–primer binding and prevent influenza transcription. In addition, the conserved regions of PB2 are expected to be less prone to mutation compared with surface proteins such as HA or NA [12]. Therefore, PB2 is considered an ideal target for the development of novel anti-influenza drugs [2,13,14].

Pimodivir (VX-787, JNJ872) is an influenza virus cap-binding domain (CBD) inhibitor being developed by Vertex Pharmaceuticals. It has completed phase II clinical trials and is currently undergoing phase III clinical trials [15]. However, the poor pharmacokinetic properties derived from its structure have hindered the further development of pimodivir. Several pimodivir derivatives, such as compound **a** [16], **b** [16], **c** [16], **d** [16], and e [17], have been developed to have improved pharmacokinetic properties (Figure 1). While some progress has been made, the pharmacokinetic problem derived from the chemical structure cannot be fundamentally solved.

In addition to pimodivir and its derivatives, several other structural types of PB2 inhibitors have been reported (Figure 2). Compound RO0794238 is a guanine derivative that can selectively bind to influenza virus PB2 CBD. Its half maximum inhibitory concentration (IC50) value for influenza virus ribonucleoprotein (RNP) is 45 μmol/L [18]. Compounds g, h, and i were obtained by the structural modification of RO0794238, and their IC50 values for the cytopathic effect (CPE) of H3N2 influenza virus were 5.3 μmol/L, 0.6 μmol/L and 7.5 μmol/L, respectively [19]. Compound CSV0C019002 was a lead compound screened based on CPE, and had a median effective inhibitory concentration (EC50) value of 4.27 μmol/L for A/WSN/33 [20]. Compound BPR3P0128 was obtained by the structural modification of CSV0C019002, and had an EC50 value of 51–190 nmol/L for influenza A and B. BPR3P0128 acts on the replication process of influenza virus, and has a significant inhibitory effect on host pre-mRNA cap-dependent transcription, proving that it is a PB2 subunit inhibitor [21]. Spirostaphylotrichin X is a new type of spirostaphylotrichin γ-lactam compound derived from marine fungi. It has a significant inhibitory effect on a variety of influenza viruses, with an IC50 value ranging from 1.2–5.5 μM. Mechanism studies have shown that Spirostaphylotrichin X can inhibit the activity of polymerase PB2 protein by binding to the highly conserved region of PB2 CBD [22]. Compound D715-2441 (1,3-Dihydroxy-6*H*-benzo[c]chromen-6-one) is a PB2 inhibitor obtained through high-throughput screening. It can interact specifically with the influenza virus polymerase PB2 subunit and prevent PB2 cap and m7 GTP binding in a dose-dependent manner [23]. Although these compounds have novel structures, their anti-influenza activities are not ideal.

Thus, searching for new types of PB2 inhibitors with novel structures is of great significance for the development of anti-influenza drugs. Here, we report the discovery of novel structural types of influenza virus PB2 CBD inhibitors via a combination of in silico structure-based virtual screening approaches and biological evaluations.

## 2. Results

### 2.1. Virtual Screening

#### 2.1.1. Receptor Selection

Among the reported structures of proteins containing active ligands, only three proteins had a complete CBD structure (PDB: 5WL0, 6EUY, 6EUV). Our research team has used 5WL0 (resolution: 1.27 Å) for virtual screening (to be published on Bioorg Med Chem), so in this study, protein 6EUV with a higher resolution among the remaining two proteins, 6EUV (resolution: 2.7 Å) [24] and 6EUY (resolution: 3.0 Å), was selected for virtual screening.

#### 2.1.2. Verification of the Docking Method Preparation

To investigate the applicability and reliability of the docking method, we first attempted to redock ligand VX-787 to its binding site in the crystal structure using the Glide module (Schrodinger suite). The unit cell of the crystal structure of influenza A H3N2 PB2 (241–536) protein combined with VX-787 (PDB: 6EUV) contains four independent chains, namely chains A, C, F and I. Each chain is an independent PB2 CBD protein bound by VX-787. Each chain was prepared using the protein preparation wizard module (Schrodinger suite), and their Glide grid was generated using the receptor grid generation module (Schrodinger suite), as described in the experimental section. VX-787 was extracted from the protein, and was prepared using the LigPrep module (Schrodinger suite). Glide docking was performed with XP precision using the grid files generated by chains A, C, F and I, respectively. Ligand (VX-787) was set as flexible as the default parameter. Post-docking minimization was also performed, and the docking results for each protein chain were analyzed. It was found that the XP Gscores of the docking poses of VX-787 with Chain I were the highest, and the highest-ranked pose (XP Gscore: −10.440) has the smallest root mean square deviation (RMSD) from the original pose (0.28 Å) (Appendix A). This result suggested that the docking procedure is able to accurately predict the binding pose of the true substrate, and is likely capable of selecting active compounds from a pool of chemicals, and Chain I of 6EUV is the most suitable chain for docking research.

#### 2.1.3. Virtual Screening, ADMET Property Prediction and Compound Selection

An internal compounds library consisting of 8141 molecules was processed and subjected to Glide virtual screening workflow (VSW) module (Schrödinger suite) processing to obtain the initial hits. The Glide grid file and docking parameters used during docking are the same as those used in the docking method verification section. The Glide-VSW is a hierarchical multi-precision docking protocol involving three levels of increasing docking precisions: high-throughput virtual screening (HTVS); standard precision (SP); and extra-precision (XP). The top 20% of the HTVS docking outputs and SP docking outputs were applied in the next screening step. The top 30% of the Glide-XP docking outputs retrieved 79 molecules, from which 65 compounds with XP GScore below −7.0 were filtered for final analysis (Appendix A).

In addition to examining the XP GScores, we also evaluated the ADME properties of each compound using the ADMET protocol in Discovery Studio 3.0 (DS3.0), focusing on solubility and absorbability, the two most important properties affecting activities. The solubility of the compound was evaluated using the “ADMET Aqueous Solubility Level” predictor, and the absorbability was evaluated using the “ADMET Absorption Level” predictor. To avoid the aggregation of selected compounds in certain specific structure types, these 65 molecules were divided into 25 clusters according to their structural fingerprint (FCFP_6). In each cluster, those compounds with low scores in XP GScore and with ADMET absorption and ADMET water solubility levels that met the requirements for drug development were selected. Finally, 16 compounds (Appendix A) were selected for activity evaluation. The selection rate of virtual screening was approximately 0.20%.

### 2.2. Biochemical Assays

#### 2.2.1. Cytopathic Effect (CPE) Inhibition Assay and Cytotoxicity Assay

The antiviral activities of the candidate compounds against influenza A/Puerto Rico/8/1934 (H1N1) and HK/68 (H3N2) viruses were determined in Madin–Darby canine kidney (MDCK) cells via CPE inhibition assays. Oseltamivir carboxylate (OC) and VX-787 were used as positive controls. Six compounds (Str1614, Str1916, Str3107, Str5776, Str6318, Str7374) were found to rescue cells from both H1N1 and H3N2 virus-induced CPEs, with EC50 (the concentration for 50% of maximal effect) values ranging from 2.51 μM to 55.43 μM at non-cytotoxic concentrations (Figure 3, Table 1). There was no significant difference in the antiviral activity of each compound against the two H1N1 and H3N2 virus subtypes. The hit rate of biological screening compounds was close to 1/3. The most active compound, Str5776, had EC50 values of 3.55 ± 1.23 μM and 2.51 ± 0.12 μM for H1N1 and H3N2 influenza viruses, respectively. The EC50 values of compound Str1916 against H1N1 and H3N2 influenza viruses were 4.82 ± 3.03 μM and 16.97 ± 7.94 μM, respectively. The 50% cytotoxic concentration (CC50) values of these six compounds were all greater than 100 μM.

#### 2.2.2. Surface Plasmon Resonance (SPR) Analysis

To confirm that the six active compounds are competitive ligands of PB2, a biomolecular interaction analysis based on an SPR competitive binding assay was conducted (compound Str3107 was not assayed because of insufficient quantity). Consistent with our expectations, which were based on previous molecular docking experiments, moderate to good binding affinities with equilibrium dissociation constants (Kd), ranging from 0.081 μM to 1.53 μM, were detected between these compounds and influenza PB2 CBD (Table 1). Compound Str1614 showed concentration-dependent binding and a dissociation mode with a Kd of 0.081 μM, which is close to the Kd value of VX-787 (0.054 μM). These results indicated that our compounds have good binding affinities for the influenza virus PB2 CBD.

Compared with reported PB2 inhibitors (Figure 1 and Figure 2), we found that the active compounds that we verified had novel chemical scaffolds. In addition, these compounds had relatively simple structures and were easy to synthesize, so they were suitable for optimization as lead compounds for influenza virus inhibition. It is worth mentioning that, despite the Kd value of compound Str1614 being close to that of VX-787, the antiviral activity at the cellular level was not high. It is speculated that this may be related to the low solubility and/or its low permeability of Str1614. Its predicted that ADMET Absorption Level value was 2 (low), and its predicted QPPMDCK value was 27.3 nm/s (close to the forbidden value of 25 nm/s). It may be possible to find a new structural type of highly active PB2 inhibitor by optimizing Str1614.

### 2.3. Molecular Dynamics (MD) Simulation and Analysis

#### 2.3.1. MD Simulation Using Desmond

To analyze the interactions between influenza PB2 and our active compounds, the structural complexes of influenza PB2 docked with VX-787 and our active compounds were assessed by MD simulation using Desmond [25]. Considering the trade-off between simulation accuracy and computing power, the MD simulation was executed for 200 ns. The RMSD of PB2 protein Cα atoms and Lig fit Prot of ligands in the MD simulations is presented in Appendix A. The results showed that the RMSD of PB2 protein Cα atoms tended to be stable during the simulation process, and there was no case in which the Lig fit Prot value was significantly greater than the corresponding protein RMSD value (Appendix A). This confirmed that the molecular dynamics simulation results were reliable, and the ligand did not diffuse away from its initial binding site. The root-mean-square fluctuations (RMSF) of amino acid residues in the MD simulations are presented in Appendix A. We can see that, except for the *N*-terminus and C-terminus of the protein, the amino acid residues with large fluctuations were concentrated in a few fragments, and except for a small number of amino acid residues, the RMSF of most amino acid residues in all complexes was less than 3.0 Å. These results further indicated that the protein was stable during the simulation.

The stabilities of the PB2–ligand complexes were further evaluated by protein-ligand contacts and protein-ligand interaction fractions. Protein–ligand contacts (or ‘interactions’) are categorized into four types: hydrogen bonds (HB), hydrophobic contacts (HP), ionic bridges and water bridges (WB). Each interaction type contains more specific subtypes, for example, hydrophobic contacts fall into three subtypes: ion-pi interactions (ion-pi), pi-pi stacking (pi-pi) and non-specific hydrophobic interactions (ns HP) [26]; Figure 4A–G presents a 2D summary of the protein–ligand contact analysis results for PB2 binding with VX-787 or the active compounds (Str1614, Str1916, Str3107, Str5776, Str6318, Str7374). Protein–ligand interactions, such as HB, WB, ion-pi interactions and pi-pi stacking, occurring with a probability of over 30% during the simulation, are displayed. Figure 5A–G presents the protein-ligand interaction fractions of the possible four types of bond interactions (HB, HP, ionic and WB), for PB2 binding with VX-787 or the active compounds. The interaction fraction of the protein with the corresponding residues is represented by a standardized stacked bar graph; for example, a value of 0.7 means that a specific interaction is maintained for 70% of the simulation time. Values above 1.0 are possible, because certain protein residues may form multiple contacts of the same subtype with the ligand.

#### 2.3.2. Representative Structures of the PB2 CBD/Ligand Complexes and Binding Free Energy Analysis by MM/GBSA

We conducted a trajectory clustering analysis using the “RMSD Based Clustering of Frames from Desmond Clustering Trajectory” in Maestro to estimate the most populated representative structure in each MD simulation. The structure with the most neighbors in the structural cluster was selected as the representative structure for each complex. For a detailed view of the representative structures, see Figure 6A–G.

The binding free energy (ΔG bind) of each representative structure was calculated using the generalized born surface area (MM/GBSA) method, and the results are listed in Table 1. The results confirmed that the binding of our active compounds to PB2 was relatively strong. The ΔG bind of these representative conformations of PB2/our active compounds complexes ranged from −55.37 kcal/mol to −87.36 kcal/mol. In contrast, the ΔG bind of the representative conformation of the PB2/VX-787 complex was −85.68 kcal/mol. However, we also found that the correlation between the calculated ΔG bind and the equilibrium dissociation constant (Kd) determined by the SPR method was not obvious. This may be owing to the following reasons: first, the MMGBSA method was used to measure the thermodynamic stability of the model, which cannot consider dynamic stability, such as the energy barrier that occurs when the ligand binds to PB2, while the equilibrium dissociation constant (Kd) measures a kinetic process. Second, owing to the expensive computational cost and low predictive accuracy, a large deviation remains between the results obtained by our dynamic simulation and the real world. In addition, the equilibrium dissociation constant (Kd) determined by the SPR method is affected by many environmental factors, and does not completely represent the real world.

## 3. Discussion

### 3.1. The Screening Workflow

In this study, we developed a workflow for identifying hit compounds targeting viral proteins by a combination of in silico structure-based virtual screening approaches and biological evaluations. Specifically, 8417 small molecules from our internal compound database were docked to the PB2 cap-binding site with the Glide VSW, using the Chain I of PDB 6EUV as a receptor protein. Their predicted binding scores were ranked, and their predicted binding poses were inspected. The 65 output compounds were selected on the basis of XP G Score, solubility, absorbance and structural fingerprint, and finally, 16 compounds were selected for biological evaluation. The selection rate of virtual screening was approximately 0.20%. The antiviral activity of the compounds against A/Puerto Rico/8/1934 (H1N1) and HK/68 (H3N2) influenza virus was confirmed in MDCK cells by a CPE inhibition test. SPR was used to determine the binding affinity of the active compound to H1N1 influenza virus PB2. Six compounds were found to rescue H1N1 and H3N2 virus-induced cytoses at non-cytotoxic concentrations, with EC50 values ranging from 2.51–55.43 μM. The hit rate of biological screening compounds was close to 1/3. Moreover, an SPR test showed that the active compound had good affinity for the H1N1 and H3N2 PB2 CBD. This workflow enabled us to identify virtual hits through in silico virtual screening, with high reliability and a high hit rate. Next, we conducted MD simulation studies on the docking complex of these active compounds. A stepwise flow diagram of the present work is illustrated in Figure 7.

### 3.2. Analysis of PB2-Ligand Binding Model Based on Dynamic Simulation Results

Figure 4A–G and Figure 5A–G show 2D summaries of the interaction analysis results and the protein–ligand interaction fraction results from the MD simulation of the influenza PB2–ligand complexes. Figure 4A–G illustrates the representative structure, with the largest population in the MD simulation of the PB2–ligand complexes. Figure 4A–G and Figure 5A–G show the average state of the protein–ligand complexes over a period of simulation time, while Figure 6A–G reports the conformations of different protein–ligand complexes with the highest probability of occurrence during the simulation. Although the states described in Figure 4A–G, Figure 5A–G and Figure 6A–G are not the same, the close inspection of the key interactions between ligands and proteins revealed that they are consistent with each other, and can be mutually confirmed. On the basis of the protein–ligand interaction information provided by MD simulations, we analyzed the PB2–ligand binding model.

The information derived from the MD simulations of the PB2–VX-787 complex (as can be seen in Figure 4A, Figure 5A and Figure 6A) suggested that the pyrimidinyl-azaindole fragment of VX-787 interacted with the protein residues Lys376 and Glu361 via HB and with His357, Phe323, Phe363 and Phe404 side chains via pi-pi stacking. The carboxylic group of VX-787 participated in three water-mediated interactions at His357 and Asn510, as well as the main chain carbonyl of Arg355. In addition, Val511 maintained a hydrogen bond and water bridge, with the NH linking pyrimidinyl-azaindole and bicyclo [2.2.2]octane (Figure 4A). The information on the interaction of PB2 and VX-787 is consistent with the crystal structures of the PB2/VX-787 complexes reported in the literature and the binding mode information for VX-787 and PB2 [27].

Among our six compounds, the structures of compounds Str3107 and Str5776 were similar, both possessing two aromatic acids connected by a methylene group. The difference is that the aromatic acid of Str5776 is benzoic acid, and the aromatic acid of Str3107 is naphthoic acid (Figure 4D,E). Among the two carboxyl groups of Str3107 and Str5776, one carboxyl group (carboxyl group 1) forms a strong hydrogen bond with Arg355, and the other carboxyl group (carboxyl group 2) and the adjacent phenolic hydroxyl group form a network of HB and WB with Arg332, Phe404 and Lys376. The carboxyl group 2 of Str5776 also forms a hydrogen bond and water bridge with Arg508, and the carboxyl group 2 of Str3107 forms a water bridge with Glu361 (Figure 4D,E and Figure 5D,E). However, the orientation of aromatic groups differs (Figure 6D,E). The benzene ring on one side of Str5776 can form pi–pi stacking with His357 and a non-specific hydrophobic interaction with Phe404. The benzene ring on the other side has a hydrophobic effect on Met431. Meanwhile, the two naphthalene rings of Str3107 can only form non-specific hydrophobic interactions with Phe323 and Val511, and cannot form pi–pi stacking with PB2 (Figure 4D,E and Figure 5D,E). The other three compounds, Str1916, Str6318 and Str7374, all contain only one carboxyl group, but there are significant differences in structure between the three (Figure 4C,F,G). Str6318 has a relatively simple structure. The carboxyl group of its o-acylbenzoic acid fragment can simultaneously form HB, WB and salt bridges with two basic amino acids, Arg355 and Lys339. The carbonyl group of its o-acylbenzoic acid can form HB with Val511. The benzene ring on the other side of Str6318 can form a pi–pi stack with Phe323. Meanwhile, no other important interactions between Str6318 and PB2 protein have been observed (Figure 4F, Figure 5F and Figure 6F). The entire molecule of compound Str1916 is linearly arranged (Figure 4C). When compound Str1916 binds to PB2 protein, the entire molecule folds into a V shape (Figure 6C). The thiazolyl amino group is located at the bottom of the ‘V’, and forms a strong hydrogen bond with Asn429, and the amino group of thiazole can form a water bridge with Lys339. The carboxyl group of compound Str1916 has a large degree of freedom. It can simultaneously form HB, salt bridges and WB with the basic amino acids Arg355 and Lys399, and can also form WB with Lys353 and Arg508. The benzene ring connected to the 4-position of the thiazole of compound Str1916 can form a pi–pi stack with Phe323 and Phe404 (Figure 4C, Figure 5C and Figure 6C). The structure of Str7374 is relatively complex, and can be regarded as the condensation product of an aromatic acid, with a complex aromatic ring and phenylalanine (Figure 4G). The formylphenylalanine of Str7374 contributes most to its binding to PB2, and its carboxyl and carbonyl groups can form complex HB and WB networks with Arg355 and His357. The amino group of the three-membered ring forms HB with Gln406 and Arg355, and the carbonyl group forms HB with Lys376. The three-membered aromatic ring forms a non-specific hydrophobic interaction with Phe404. The phenyl ring region of phenylalanine forms hydrophobic interactions with Mett431 and Pro430 (Figure 4G, Figure 5G and Figure 6G). Compound Str1614 is the only compound among our six compounds that does not contain a carboxyl group (Figure 4B). The thiazole nucleus and the p-hydroxyphenyl group at the 4-position of thiazole form pi–pi stacking with Phe323, Phe363 and His357, and HB with Glu361 and Gln406. The trihydroxybenzamide fragment connected to the 2-position of the thiazole nucleus can form a strong hydrogen bond with Val511, while the *N* atom of the thiazole nucleus and the carbonyl oxygen of the trihydroxybenzamide fragment form a network of HB and WB with His357. The benzene ring of the phenylethyl group at the 5-position of the thiazole nucleus forms an ion–pi interaction with Arg332, which is a unique interaction between compound Str1614 and PB2. In addition, compound Str1614 can also form a strong hydrophobic interaction with Phe404 (Figure 4B, Figure 5B and Figure 6B).

Through the above analysis of the MD simulation results of the PB2–ligand complexes, we revealed the unique binding modes of our active compounds that are distinct from the binding mode of VX-787. This is consistent with the greater structural diversity observed for the six compounds. However, the careful inspection of the MD simulation results for the PB2–ligand complex confirmed that our compounds still share common features in terms of protein binding. As expected, the amino acids that play a key role in maintaining the binding of our compound or VX-787 to ligand are highly conserved. To investigate this further, we used MD simulation to analyze the key amino acid residues that maintain the binding of the ligand to the PB2 protein.

### 3.3. Key Amino Acid Residues of PB2 CBD Analysis Based on Dynamic Simulation Results

To determine the amino acid residues that play a key role in the binding of protein and ligands, according to the information provided in Figure 5A–F, Table 2 provides a semi-quantitative summary of the protein–ligand interaction fractions between PB2 CBD and VX-787 of our active compounds. The protein–ligand interaction fractions between different ligands and different amino acid residues, semi-quantitatively expressed as dash (-), dot (•), asterisk (*) and multiple asterisk, can be read from the corresponding cell (the intersection of column and row). The cumulative value of the interaction fractions between the different residues and our six active compounds is shown in the last column.

It can be seen from Table 2 that, although the binding modes of the six active compounds that we found differ, the amino acids that play a key role in maintaining the compound and protein binding are highly conserved. The amino acid residues that play a key role in maintaining binding to the protein include: Phe323, Phe325, Arg332, Lys339, Arg355, His357, Glu361, Lys376, Phe404, Glu406, Asn329, Met431, Arg508, Asn510 and Val511 (16 residues, >3 asterisks), of which 12 amino acid residues play a role in maintaining the binding of VX-787 to the protein, and only four amino acid residues (Phe325, Arg332, Glu406, Arg508) are not involved in the binding of VX-787 to the PB2 protein. The amino acid residues that play an important role in maintaining their binding to the protein are Phe323, Arg332, Lys339, Arg355, His357, Lys376, Phe404, Glu406, Asn329, Arg508 and Val511 (11 residues, >5 asterisks), of which eight amino acid residues play an important role in maintaining the binding of VX-787 to the protein, and only three amino acid residues (Arg332, Glu406 and Arg508) are not involved in the binding of VX-787 to the PB2 protein. The amino acid residues that play a key role in maintaining their binding to the protein are Phe323, Arg332, Lys339, Arg355, His357, Phe404, Arg508 and Val511 (8 residues, >7 asterisks). Among them, four amino acid residues (Phe323, Arg355, His357 and Val511) play a key role in maintaining the binding of VX-787 to the protein (>3 asterisks), and two amino acid residues (Lys339, Phe404) play a secondary role in maintaining the binding of VX-787 to protein (>3 asterisks). Only two amino acid residues (Arg332 and Arg508) do not play an important role in maintaining the binding of VX-787 to protein.

It is worth mentioning that, among these amino acid residues, Arg355 plays an important role in the binding of VX-787 or our active compound to PB2 [24,27,28]. The guanidine group of Arg355 can form multiple HB with VX-787 and the carboxyl group of our compound, and can also form WB and ion interactions (Figure 4A,C–G and Figure 5A,C–G, Table 2). The skeleton carbonyl group of Arg355 can also form strong HB with compound Str7374 (Figure 5G). Another important amino acid residue is His357, which can form pi–pi stacking, HB and WB with VX-787 [24,27,28], and our active compounds (Figure 5A–G, Table 2). Phe404 and Phe323 also play an important role in maintaining the binding of our compound and VX-787 to proteins [27]. The main form of interaction between Phe323 and ligand is hydrophobic interactions, including non-specific hydrophobic interactions and pi–pi stacking (Figure 4A–D,F and Figure 5A–G, Table 2). Unlike Phe323, Phe404 can form a water bridge with Str3107 and Str5776, in addition to non-specific hydrophobic interactions and pi–pi stacking (Figure 4A–G and Figure 5A–G, Table 2). Lys339 can form HB, WB and ion bridges with our compounds (Figure 4A,C–G and Figure 5A–G, Table 2). Val511 is another important amino acid residue. In addition to forming HB with ligands, it can also form strong non-specific hydrophobic interactions (Figure 4A–G and Figure 5A–G, Table 2).

The other three amino acid residues worthy of attention are Arg332, Arg508 and Gln406. These two amino acid residues play a key role in maintaining the binding of our active compounds to PB2 protein, but they do not participate in the binding of VX-787 to the protein. Arg332 interacts with the phenylethyl group of Str1614 at the 5-position of the thiazole nucleus to form an ion–pi interaction, a water bridge, a hydrogen bond, and an ion bridge with carboxyl group 2 of Str3107, and forms a hydrogen bond and water bridge with carboxyl group 2 of Str5776. Arg508 forms HB and WB with carboxyl group 2 of Str5776, and forms WB, HB, and ionic bridges with Str1916. Gln406 can form strong HB with Str1614, WB and non-specific hydrophobic interactions with Str3107, and WB and HB with Str5776.

The above results indicate that, although the six active compounds obtained in this study are structurally different to VX-787 and their binding mode with PB2 protein is also distinct, the amino acid residues that play a key role in protein binding are highly conserved between our compounds and VX-787. This shows that, when the ligand binds to PB2 protein, regardless of the structure of the ligand, the dominant amino acid residues that play a key role are relatively fixed. Dominant amino acid residues play a more important role in ligand binding than other amino acid residues. Highly active PB2 inhibitors thereby need to interact with these dominant amino acid residues as fully as possible. The simulation results show that VX-787 can be closely integrated with PB2. VX-787 can interact with multiple dominant amino acid residues, such as Phe323, Arg355, His357, Glu361 and Val 511, and often forms multiple interactions [27]. This is consistent with the high affinity of VX-787 toward PB2 and its significant antiviral activity [29]. By contrast, the possible interactions between our active compounds and these dominant amino acid residues were significantly fewer than those of VX-787. To improve the activity of our compounds, we must improve their adaptability to the PBCBD site, and increase the number and intensity of interactions with dominant amino acid residues. The above simulation results also suggested that Arg332, Arg508 and Gln406 are important amino acid residues in the PB2 binding site, but they are not involved in the binding of VX-787 to the protein. In future studies, increasing the interaction with these dominant amino acid residues should be considered as a strategy to optimize the activity of PB2 inhibitors.

## 4. Materials and Methods

### 4.1. Virtual Screening

#### 4.1.1. Receptor Preparation

The crystal structure (PDB: 6EUV) of influenza A H3N2 PB2 (247–536) bound to VX-787 was downloaded from the protein data bank. The raw PBD protein structure was prepared by using the Protein Preparation Wizard (Schrodinger), adding hydrogen atoms, refining the loop region, optimizing H-bond assignment, and finally, restrained energy minimization (hydrogens only) by using an OPLS-2005 force field.

#### 4.1.2. Glide-Grid Generation

The Glide-grid was generated using the Receptor Grid Generation module. The site for docking analysis was generated using the structural coordinates of the co-crystallized ligand VX-787. The center of VX-787 (-24.06(X), 6.77(Y) 10.63(Z)) was designated as the grid center, the innerbox was set to 10′10′10 (angStroms), the outerbox was set to 23.21′23.21′23.21 (angStroms). No water molecules remain in the protein, and no constraints, no rotatable groups, or exclude volume were set.

#### 4.1.3. Ligand Preparation

An internal database consisting of 8141 molecules was subjected to the LigPrep module to apply forth field (OPLS-2005z), generate ionization states at pH 7.0 ± 2.0, and multiple conformers to develop a new database. This new database was used for the docking-based virtual screening process.

#### 4.1.4. Structure-Based Virtual Screening of Compound Libraries

Molecular docking was performed using Glide module (Schrodinger). Databases were subjected to a virtual screening workflow (VSW) module to obtain the initial hits. The Glide-grid generated as above mentioned was used as a receptor grid. Firstly, all molecules were screened using Glide HTVS mode. The top 20% of the HTVS outputs were screened using the SP docking mode. Secondly, the top 20% molecules from SP docking were processing through XP docking mode. Finally, the top 30% of the output of XP docking was retained. Compounds with scores below −7.0 enter the next round of selection.

#### 4.1.5. In Silico ADME Analysis and Compound Selection

The ADMET protocol in Discovery Studio 3.0 (DS3.0) (Accelrys, USA) was used to calculate the ADMET predictors for each selected compound. The solubility of the compound was evaluated using the “ADMET Aqueous Solubility Level” predictor, and the absorbability was evaluated using the “ADMET Absorption Level” predictor. According to the structural fingerprint (FCFP_6), these 65 molecules were divided into 25 clusters using the “Cluster Ligands” protocol of DS3.0. In each cluster, the compound with an ADMET Absorption Level of 0 (good) or 1 (moderate); ADMET Aqueous Solubility level of 2 (Yes, low), 3 (Yes, good), or 4 (Yes, optimal); and the lowest XP GScore was chosen. If there is no compound that meets the requirements in the cluster, but the XP GScores of compounds in this cluster are generally high, then a compound is also selected. At the same time, if there are more compounds in this cluster, more compounds can be selected in this cluster according to the XP GScore. In total, 16 compounds were selected for activity evaluation. The structure, ID, XP GScore, cluster and ADMET predictors of the selected compounds are shown in Appendix A.

### 4.2. Biological Evaluation

#### 4.2.1. Cells and Viruses

Madin–Darby canine kidney (MDCK) cell line was purchased from the American Type Culture Collection (ATCC, Manassas, VA) and maintained at 37 °C in Dulbecco’s modified Eagle’s medium (DMEM) supplemented with 10% fetal bovine serum, 100 μg/mL of Streptomycin and 100 U/mL of penicillin. Influenza A/Puerto Rico/8/1934 (H1N1) (PR/8), A/LiaoNing-ZhenXing/1109/2010 (ZX/1109 H1N1, oseltamivir-resistant isolate, provided by Dr. Yuhuan Li, CAMS), PR/8-R292K mutant (H1N1, oseltamivir-resistant, recombinant Strain), PR/8-I38T mutant (H1N1, baloxavir-resistant, recombinant Strain), A/WSN/33 (H1N1), A/Hongkong/8/68 (H3N2) (HK/68) and influenza B/Lee/40 were propagated in 8–10 day old embryonated chicken eggs or MDCK cells for 3 days at 37 °C. The virus strains were stored in our lab.

#### 4.2.2. CPE Inhibition Assay

MDCK cells were seeded and grown for 18–24 h to a confluent monolayer in 96-well plate. The medium was changed to DF-12 medium containing 2 μg/mL TPCK-trypsin before PBS washing twice. Cells were infected with influenza virus at an MOI of 0.005 in DF-12 medium containing 2 μg/mL TPCK-trypsin in the presence of various concentrations (ranging from 0.005 μM to 100 μM by a three-fold dilution) of the test compound. After 72 h incubation at 37 °C in CO_2_ incubator, the antiviral activity of test compounds was measured using Celltiter-Glo viability assay (Promega). The concentration for 50% maximal effect (EC50) was calculated by Origin 8 software.

#### 4.2.3. Cytotoxicity Assay

The cytotoxicity of compounds was evaluated in MDCK cells using Celltiter-Glo viability assay following the kit manual. Briefly, cells were seeded at a density of 1.5 × 104 per well into 96-well plates, and grown for 18–24 h to a confluent monolayer. The medium was changed into DF-12 medium containing 2 μg/mL TPCK-trypsin, and the test compounds were added to cells by a three-fold dilution series. DMSO was added as control. After 72 h incubation at 37 °C in CO_2_ incubator, the luminescence was read by a SpectraMax M5 microplate reader (Molecular Device). The 50% cytotoxicity concentration (CC50) was calculated by Origin 8 software.

#### 4.2.4. Surface Plasmon Resonance (SPR) Analysis

HIN1 PB2 CBD protein expression and purification: 507bp of target gene was ampified by primers 190822-F2 and 190822-R1 from plasmid spET24-6H-JS30-2. Then, the 507bp fragment was used as the template to amplify the 531bp by using primers 190822-F3 + 190822-R1 for the purpose of adding 3C protease sequence. Finally, the 531bp was cloned to vector spET24-SUMOstar (StuI + XhoI) by homologous recombinantion with *N*-his and sumo tag. The protein was expressed in the Escherichia coli expression system. Then, the fusion protein was captured by NI-IDA column, the tag was removed by cleavage with 3C enzyme, and the digested protein mixtures were further purified by NI-IDA column, giving the target protein PB2 CBD of influenza A/WSN/1933(H1N1) (aa. 318–486).

H1N1 PB2 was immobilized to a CM5 sensorchip (GE Healthcare) to a level ~ 4900 response units (RUs) using a Biacore T200 (GE Healthcare) and a running buffer composed of 1 × PBS-P + 0.02 M phosphate buffer, 2.7 mM KCl, 0.137 M NaCl and 0.05% Tween 20. Serial dilutions of small molecules were injected, ranging in concentration from 20 to 0.156 µM. The resulting data were fit to the affinity binding model using Biacore Evaluation Software (GE Healthcare).

### 4.3. Molecular Dynamics Simulation and Analysis

#### 4.3.1. Molecular Dynamics Simulation

MD simulations for interaction analysis were performed using Desmond [30]. The simulation systems were set up using the System Builder in Maestro. The PB2/ligand complex models were placed in the orthorhombic box at a buffer distance of 10 Å to create a hydration model. An SCP water model [31] was used to create the hydration model. The cut-off radii for van der Waals and electrostatic interactions, time step, and the initial temperature and pressure of the system were set to 9 Å, 2.0 fs, 300 K, and 1.01325 bar, respectively. The sampling interval during the simulation was set to 100 ps. Finally, MD simulations were performed under the NPT ensemble for 200 ns using an OPLS3e force field [32].

#### 4.3.2. Interactions Analysis and Trajectory Clustering for MM/GBSA

To analyze the MD simulations, the “Simulation Interactions Diagram” tool in Maestro was used to perform an interaction analysis between PB2 and the ligands, and the Desmond trajectory clustering tool was used to obtain representative structures. In the trajectory clustering, a backbone-atom was set for the RMSD matrix. “Trajectory Frame Clustering” in Maestro was used to estimate the most populated representative structure for each MD simulation. The trajectory frame extraction interval was 10 frames, 1000 frames were used for clustering each trajectory, and the maximum output number of clusters was set to 10. The structure with the largest number of neighbors in the structural cluster was used as the representative structure for binding free energy calculation. Calculation of the binding free energy was conducted by the Prime MMGBSA tool in Maestro. The VSGB solvation model and OPLS3e force field were set for binding free energy calculation.

## 5. Conclusions

In this study, we developed a workflow for identifying hit compounds targeting influenza virus PB2 proteins by a combination of in silico structure-based virtual screening approaches and biological evaluations. From an internal database containing 8417 molecules, six compounds were identified that rescue cells from H1N1 virus-mediated death at non-cytotoxic concentrations with EC50 values ranging from 2.51–55.43 μM. These compounds bound to the PB2 CBD of H1N1 with Kd values ranging from 0.081–1.53 μM. Molecular dynamics (MD) simulations analysis showed that the binding method of each of our active compounds has its own characteristics, and is also very different from the binding method of VX-787. Molecular dynamics (MD) simulations also helped us identify the dominant amino acid residues that play a key role in the binding of PB2 protein and ligands, which provides key information for the design of new inhibitors.

## Figures and Tables

**Figure 1 molecules-26-06944-f001:**
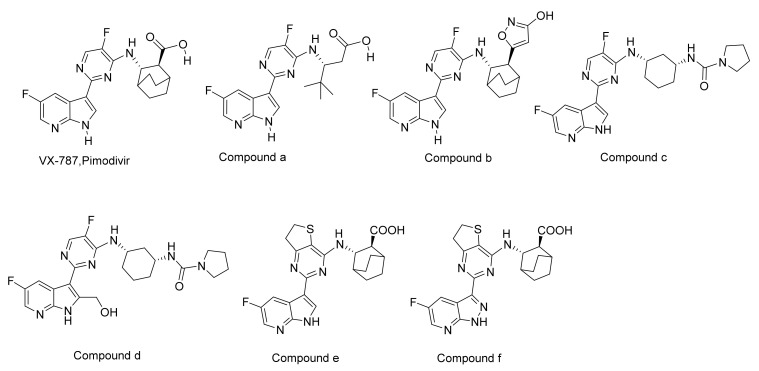
The structure of pimodivir and its derivatives.

**Figure 2 molecules-26-06944-f002:**
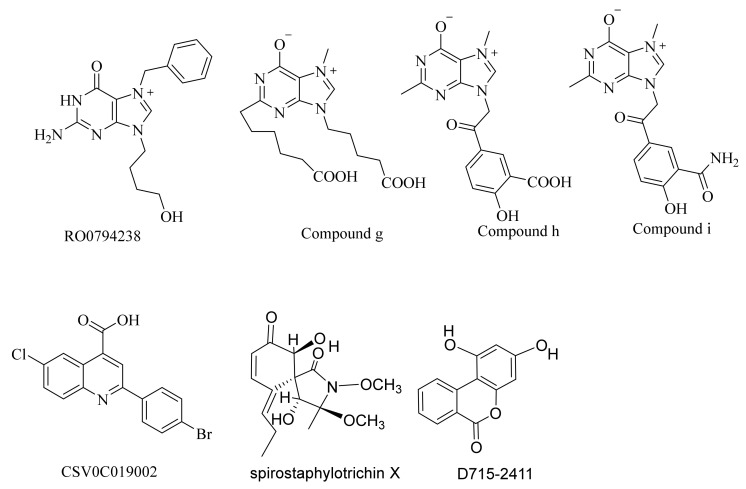
The structure of PB2 inhibitors of other structure types.

**Figure 3 molecules-26-06944-f003:**
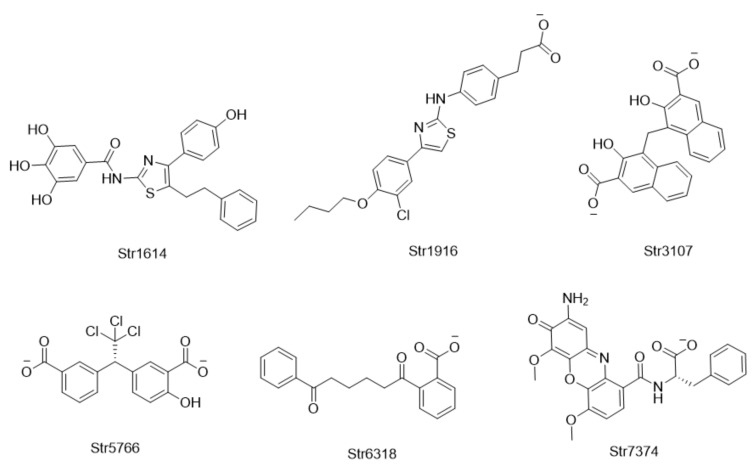
The structures and IDs of the active compounds.

**Figure 4 molecules-26-06944-f004:**
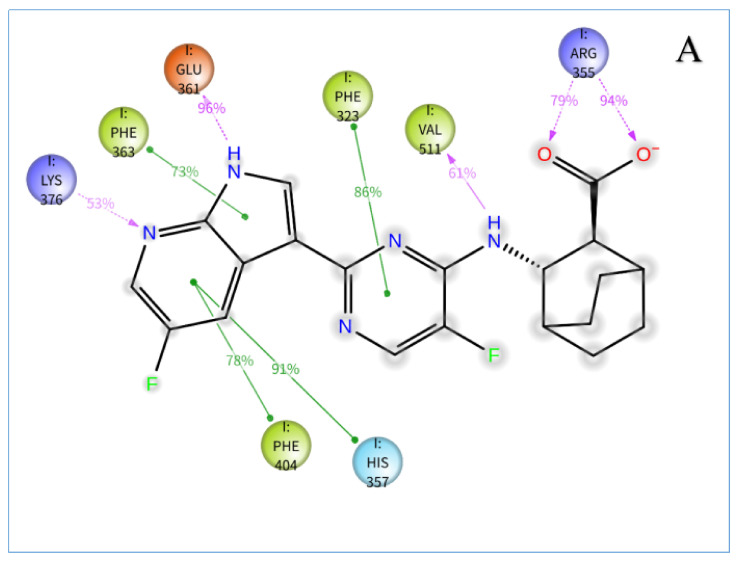
2D summary of interaction analysis results of PB2-ligand complexes. The interaction pairs that occurred during more than 30% of the simulation time are included. Shown are interactions between (**A**) VX-787 and PB2; (**B**) Str1614 and PB2; (**C**) Str1916 and PB2; (**D**) Str3107 and PB2; (**E**): Str5776 and PB2; (**F**) Str6318 and PB2; and (**G**) Str7374 and PB2.

**Figure 5 molecules-26-06944-f005:**
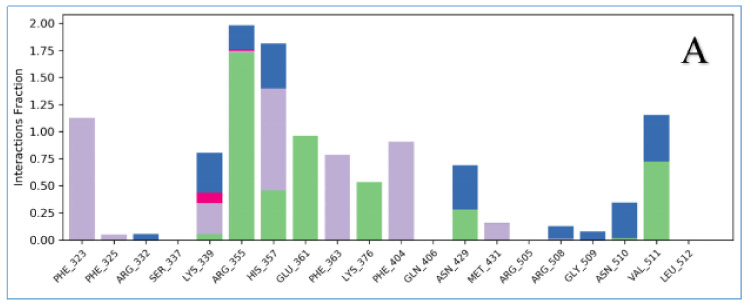
Interaction fraction summary of PB2–ligand contacts. Graph is normalized for the total simulation time. Interaction-fraction values over 1.0 indicate that the residue has multiple contact routes for interacting with the ligand. Shown are the interaction fractions for (**A**) VX-787 with PB2; (**B**) Str1614 with PB2; (**C**) Str1916 with PB2; (**D**) Str3107 with PB2; (**E**) Str5776with PB2; (**F**) Str6318 with PB2; and (**G**) Str7374 with PB2.

**Figure 6 molecules-26-06944-f006:**
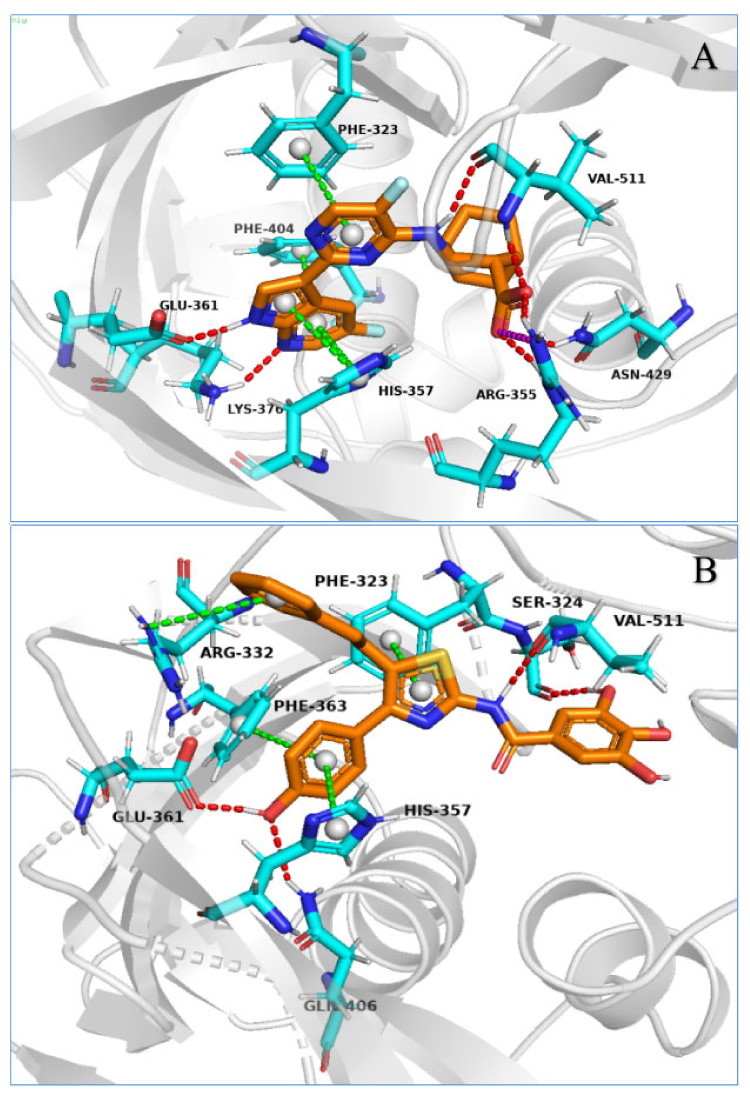
Representative structures of VX-787 or our active compounds with the largest population in the MD simulation. Shown are representative structures of (**A**) VX-787 with PB2; (**B**) Str1614 with PB2; (**C**) Str1916 with PB2; (**D**) Str3107 with PB2; (**E**) Str5776with PB2; (**F**) Str6318 with PB2; and (**G**) Str7374 with PB2.

**Figure 7 molecules-26-06944-f007:**
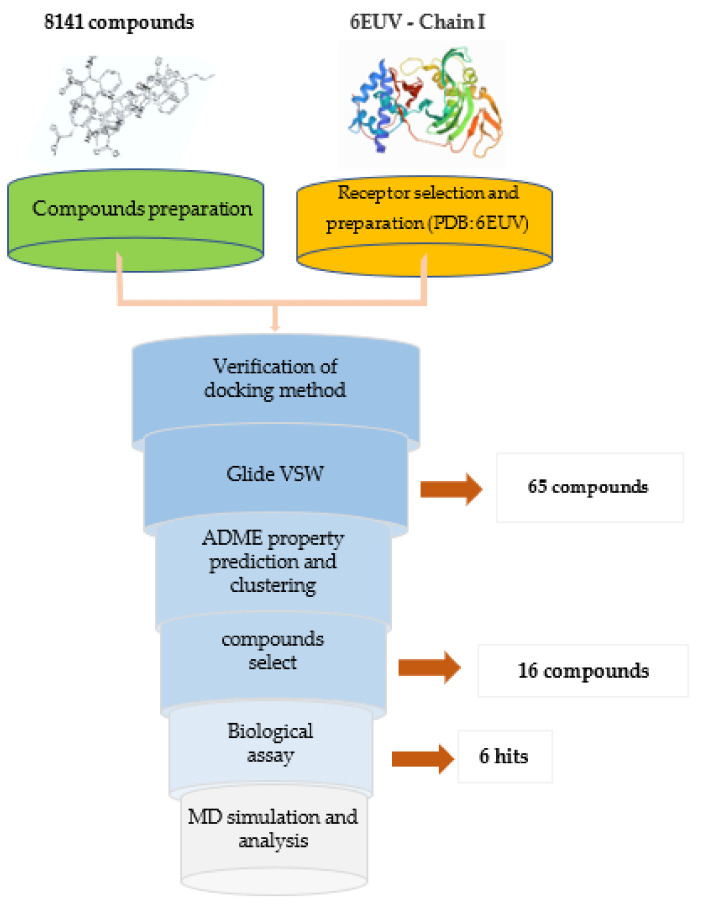
Stepwise flow diagram of the present work.

**Table 1 molecules-26-06944-t001:** Structures, docking scores, anti-H1N1 and -H3N2 activities, cytotoxicities and Kd values of the active compounds.

Compounds	Docking Score	Aqueous Solubility Level	ADMET Absorption Level	H1N1-IC50(μM)	H3N2-IC50(μM)	CC50 (μM)	SPR (Kd, (μM))	MMGBSA ΔG Bind (kcal/mol)
OC	-	-	-	0.95	0.01	>100	-	-
VX-787	−10.440	-	-	0.004 ± 0.001	0.09 ± 0.02	>100	0.054	−85.68
Str1614	−9.321	2	2	26.27 ± 8.69	19.55 ± 0.93	>100	0.081	−87.36
Str1916	−8.115	2	0	4.82 ± 3.03	16.97 ± 7.94	>100	0.917	−84.02
Str3107	−9.993	3	0	18.18 ± 4.96	45.71 ± 3.01	>100	ND *	−72.63
Str5776	−8.569	3	0	3.55 ± 1.23	2.51 ± 0.12	>100	0.178	−55.37
Str6318	−7.389	3	0	28.62 ± 9.46	11.17 ± 3.39	>100	1.530	−57.71
Str7374	−8.511	3	2	31.85 ± 8.15	55.43 ± 26.99	>100	0.910	−75.81

* ND: Not detected.

**Table 2 molecules-26-06944-t002:** Protein–ligand interaction fractions between PB2 and VX-787, or the active compounds.

	VX-787	STR1614	STR1916	Str3107	STR5776	Str6318	Str7374	The Accumulation Value of Interaction Fractions Between Residues and Our Active Compounds (Number of Asterisks).
Phe323	***	**	*	**	-	**	•	*******(7)
Ser324	-	**	-	-		•	-	**(2)
Phe325	•	*	*	*		*	•	****(4)
Arg332	•	*	•	***	****	•	-	*******(7)
Ser337	•	*	•	*	-	-	-	*(7)
Lys339	**	•	****	*	*	***	**	***********(11)
Arg355	****	•	***	******	*****	***	********	************************************(25)
His357	****	****	*	****	**	**	**	***************(15)
Glu361	**	**	-	**	-	•	•	****(4)
Phe363	**	*	•	-	**	-	•	***(3)
Lys376	*	•	-	***		-	**	*****(5)
Phe404	**	**	*	***	**	*	*	**********(10)
Glu406	-	**	-	**	*	•	*	******(6)
Asn429	**	**	**	•	•	*	•	*****(5)
Met431	*	-	*	•	*	*	*	****(4)
His432	-	*	-		-	-	-	*(1)
Arg505	-	-	•	*		•-	•	*(1)
Arg508	•	*	**	*	***	-	-	*******(7)
Asn510	*	•	*	*	•	*	*	***(3)
Vla511	***	***	*	**	•	**	*	*********(9)
Ser514	-	*				-	-	*(1)
Glu516		*						*(1)
Glu517	-	**					•	**(2)

A dash (-) indicates that the residue is not in contact with the ligand, a dot (•) indicates that the interaction fraction of the residue and the ligand is between 0 and 0.1, and an asterisk (*) indicates that the interaction fraction of the residue and the ligand is between 0.1 and 0.5. Multiple asterisks indicate that the interaction score between the residue and the ligand is greater than 0.5, and an asterisk is added for every increase of 0.5 in the interaction fraction value. The cumulative value of the interaction scores between different residues and our six active compounds are shown in the last column, represented as one asterisk or multiple asterisks.

## Data Availability

The data presented in this study are available in the references.

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
