# Peer review of "Virtual Screening and Molecular Dynamics Simulation Study of Influenza Polymerase PB2 Inhibitors"

_molecules, 2021, doi:10.3390/molecules26226944_

Round 1

Reviewer 1 Report

The abstract is poorly written and requires significant improvement.

There is no full stop in line 26.

From line 70, substantial literature is required to support the PB2 as a potent drug target with references.

Have there been other inhibitors screened against the target? The authors need to report on other in silico and wet lab work done pertaining to finding inhibitors or small molecules for PB2 cap-binding domain. They only mentioned Pimodivir.

Lines 77-78, the approach included both virtual screening and biochemical assaying, but the authors mentioned only virtual screening. This needs to be rectified.

Line 84: The authors need to cite the previous work where 5WL0 was used for virtual screening. They also did not provide the resolution for 5WL0. Why did the authors consider only resolutions and not any other criteria for choosing it? The authors need to provide the references for the papers which reported the elucidation of the protein structures.

Line 117-118: Why did the authors use top 20% of the HTVS docking outputs and top 30% of the Glide-XP docking 117 outputs? Do they have any specific reasons or literature to support such threshold or its arbitrary?

The authors apart from docking the co-crystallized ligand, they also need to use known inhibitors or drugs of the target as standards for comparison to shortlist the top hits.

Line 155- 156: “The compound Str3107 was not assayed because of insufficient quantity.” This is not sufficient reason for not undertaking the assaying of the compound. Couldn’t the authors purchase or synthesis the compound? I recommend that the compound is assayed if possible since the library of compounds were obtained in-house.

Line 153-164: The authors must provide reasons for the novelty of the chemical scaffolds of the compounds and how they can be optimized.

Line 168-169:  Why didn’t the authors find the new structural type of highly active PB2 inhibitor? At least they could have used in silico approaches to identify by doing de novo design of the subtype. I suggest that the authors do that.

Line 193-194: reference is required.

Also, portions of the manuscript are not referenced and must be addressed.

The result section contained detailed descriptions of methods and must be revised accordingly. There should be no obvious duplication between result/discussion and methods on the other hand.

Line 250: “in silico” must be in italics and must be rectified in the entire manuscript.

Line 261: The meaning of the statement is not clear: “proteins revealed that they coincide with each other”

The discussion section 3.2, numerous assertions were made without being substantiated with literature and the authors are encouraged to discuss their results within the context of previous work and highlight the novelty and implications of their study.

The screening workflow must be accompanied by a schema which shows the step-by-step methods employed to make it easy to follow the article. Sometimes it appears incoherent and must be revised entirely through the manuscript.

Line 334-340: Must be summarized as caption for Table 2. It is very difficult to follow with symbols such as asterisks. The authors must use plain English language to explain the interactions.

It appears the entire section 3.3 did not cite any literature to support the results and it appears so in other sections in the manuscript.

Line 438: “Compounds with scores below -9 enter the next round of selection.” Which scores are they referring to? The authors must be concise and precise with the writing.

Line 443: the url is wrong: (www.discoverystu dio.com); and must be corrected.

The authors must use the MM/PBSA or MM/GBSA methods to corroborate ligand-binding interactions.

Overall, the manuscript requires substantial revisions, improvement in English language and sentence structures.

Author Response

Dear Sir or Madam: 

Thanks a lot for your attention to our manuscript and we appreciate your professional and detailed criticisms and suggestions. Thank you very much for your helpful suggestion. The manuscript has been extensively revised according to your comments. For the changes made according to reviewer suggestions, we have marked them in yellow in the text. The rewritten contents have been marked in blue. The corresponding explanations of each point which is raised in your comments as follows:

The abstract is poorly written and requires significant improvement.

Response:The abstract has been rewritten to meet the requirements of the magazine as much as possible.

There is no full stop in line 26.

 Response: Thank you for your careful review, and a full stop has been added where appropriate in the text. The abstract summary part has been rewritten.

From line 70, substantial literature is required to support the PB2 as a potent drug target with references.

Response:Viral RNA polymerase is an ideal target for antiviral drugs, which has been confirmed by a large number of theories and practices. In recent years, the idea that influenza virus RNA polymer and its PB2 subunit are ideal targets for anti-influenza virus drugs has also been supported by many literatures. According to your suggestions, we have added the following three representative papers to the references:①Severin C, Rocha de Moura T, Liu Y, et al. The cap-binding site of influenza virus protein PB2 as a drug target[J]. Acta Crystallographica Section D: Structural Biology, 2016, 72(2): 245-253. ② Zhou Z, Liu T, Zhang J, et al. Influenza A virus polymerase: an attractive target for next-generation anti-influenza therapeutics[J]. Drug discovery today, 2018, 23(3): 503-518. ③ Das K, Aramini J M, Ma L C, et al. Structures of influenza A proteins and insights into antiviral drug targets[J]. Nature structural & molecular biology, 2010, 17(5): 530-538.

Have there been other inhibitors screened against the target? The authors need to report on other in silico and wet lab work done pertaining to finding inhibitors or small molecules for PB2 cap-binding domain. They only mentioned Pimodivir.

Response:We have made a detailed supplement to the research progress of pimodivir and its derivatives and the discovery of other structural types of PB2 inhibitors in the text.

Lines 77-78, the approach included both virtual screening and biochemical assaying, but the authors mentioned only virtual screening. This needs to be rectified.

 Response:Following your suggestions, the corresponding content in the paper has been revised. The corresponding content was changed to “by a combination of in silico structure-based virtual screening approaches and biological evaluations”.

Line 84: The authors need to cite the previous work where 5WL0 was used for virtual screening. They also did not provide the resolution for 5WL0. Why did the authors consider only resolutions and not any other criteria for choosing it? The authors need to provide the references for the papers which reported the elucidation of the protein structures.

Response:The work of virtual screening of PB2 inhibitors using PDB 5WL0 will be published in Biorg Med Chem in the near future. The resolution for 5WL0 has been provided in the text.

The important criteria of pdb/protein file selection to perform protein-ligand docking is the resolution and absence of gaps in the residue numbers. The three protein we mentioned here (5WL0, 6EUY and 6EUV) are all structurally complete, and there are no gaps in the number of residues. The protein with high resolution should be selected first. In the practice of virtual screening, we also found that even if the protein sequence is the same, the difference in conformation will have a great impact on the results of virtual screening. Therefore, selecting multiple proteins instead of one protein for virtual screening usually produces better results.

Line 117-118: Why did the authors use top 20% of the HTVS docking outputs and top 30% of the Glide-XP docking 117 outputs? Do they have any specific reasons or literature to support such threshold or its arbitrary?

Response:We do not have any specific reasons or literature to support such threshold. In the guide of the Glide module (Schrödinger), no guidance is given to determine the output threshold of each step of HTVS docking, Glide-SP docking and Glide-XP docking. When performing virtual screening, the threshold needs to be adjusted according to specific conditions. Initially, we specified that the output of each step of HTVS docking, Glide-SP docking and Glide-XP docking was 20%. As a result, we found that the XP GScores of these output compounds were all below -7.0, so we increased the output of the last step to 30% to avoid omissions and increase the number of candidate compounds.

The authors apart from docking the co-crystallized ligand, they also need to use known inhibitors or drugs of the target as standards for comparison to shortlist the top hits.

 Response: So far, the only known highly active PB2 inhibitors are VX-787 and its derivatives. In the section "Verification of Docking Method" of this article, we have performed the docking of VX-787, and verified the docking method accordingly. In the following discussion, we compared and analyzed the docking and molecular dynamics simulation results of the active compounds we discovered in detail with that of VX-787. I think, in this case, the scientific value of VX-787 derivative docking research is not high.

Line 155- 156: “The compound Str3107 was not assayed because of insufficient quantity.” This is not sufficient reason for not undertaking the assaying of the compound. Couldn’t the authors purchase or synthesis the compound? I recommend that the compound is assayed if possible since the library of compounds were obtained in-house.

Response:The compound Str3107 is not commercially available. Although it is selected from an internal compound library, its synthesis method is unknown to us. So, we cannot complete the synthesis and testing of this compound within the response time specified by the editor. In addition, we believe that the absence of data on the equilibrium dissociation constant (KD) of Str3107 to the protein does not have a substantial impact on the results of the paper.

Line 153-164: The authors must provide reasons for the novelty of the chemical scaffolds of the compounds and how they can be optimized.

Response:Compared with the reported PB2 inhibitors (please refer to the newly added PB2 inhibitor research background section in the introduction section), the active compounds we found have novel structures and unique binding models (See the paragraph of "Analysis of PB2-ligand binding model based on dynamic simulation results".). This is why we think the chemical scaffolds of these active compounds are novel. They can be optimized through scaffold hopping, structural modification and other methods. Due to the length of this article, we cannot systematically discuss how to optimize these six active compounds here. The purpose of this article is to discover new structural types of PB2 inhibitor lead compounds. As for how to optimize them, it is beyond the scope of this article.

Line 168-169:  Why didn’t the authors find the new structural type of highly active PB2 inhibitor? At least they could have used in silico approaches to identify by doing de novo design of the subtype. I suggest that the authors do that.

Response:The purpose of this article is to discover new lead compounds, and how to optimize them is beyond the scope of this article.

Line 193-194: reference is required.

Response:The classification criteria for protein-ligand interactions used in this article are based on Desmond User Manual 2018. The following reference has been added: “Desmond User Manual 2018 /Desmond Analyzing Desmond Simulations /Protein-Ligand Contacts from a Desmond Simulation”

Also, portions of the manuscript are not referenced and must be addressed.

Response:According to your request, some necessary references have been added to the manuscript.

The result section contained detailed descriptions of methods and must be revised accordingly. There should be no obvious duplication between result/discussion and methods on the other hand.

Response:According to your request, the result section has been rewritten.

 Line 250: “in silico” must be in italics and must be rectified in the entire manuscript.

Response:Corresponding revised have been made in the manuscript.

Line 261: The meaning of the statement is not clear: “proteins revealed that they coincide with each other”

Response:This paragraph of text has been changed to “proteins revealed that they are consistent with each other and can be mutually confirmed”

The discussion section 3.2, numerous assertions were made without being substantiated with literature and the authors are encouraged to discuss their results within the context of previous work and highlight the novelty and implications of their study.

Response:This article has added the contents of the background introduction, so that we can clarify the novelty and impact of our research.

The screening workflow must be accompanied by a schema which shows the step-by-step methods employed to make it easy to follow the article. Sometimes it appears incoherent and must be revised entirely through the manuscript.

 Response:A schematic workflow of our work has presented in the manuscript. In response to this, the content and structure of this manuscript have also been adjusted.

Line 334-340: Must be summarized as caption for Table 2. It is very difficult to follow with symbols such as asterisks. The authors must use plain English language to explain the interactions.

Response:I am sorry for my poor English. According to your suggestion, the related content of has been revised and transferred to caption for Table 2 (lines 407-412 of the revised manuscript). At the same time, in order to express the author's thoughts more clearly, we explained the meaning of the protein-ligand interaction fractions in the main text (lines 232-238 of the revised manuscript), and added instructions for Table 2(lines 399-405 of the revised manuscript). The author hopes that these will help readers understand this article.

It appears the entire section 3.3 did not cite any literature to support the results and it appears so in other sections in the manuscript.

Response:The main content of section 3.3 is to report the work of finding the dominate amino acid residues that play a key role in the binding of ligand and protein by analyzing the results of molecular dynamics simulation. This work requires little background knowledge, and there are no reports of similar work on PB2 inhibitors, so we did not cite any literature in this part of this article.

Line 438: “Compounds with scores below -9 enter the next round of selection.” Which scores are they referring to? The authors must be concise and precise with the writing.

Response:Thank you for your careful review. "Scores" in the article has been changed to "XP Gscores"

Line 443: the url is wrong: (www.discoverystu dio.com); and must be corrected.

Response:Thank you for your careful review. The url of "www.discoverystudio.com" was changed to the company name "Accelrys, USA"

The authors must use the MM/PBSA or MM/GBSA methods to corroborate ligand-binding interactions.

Response: We have used the MM/PBSA methods to calculate the free energy of binding of the representative conformation of the PB2-ligand complex and listed them in Table 1. We also discussed the calculated results in conjunction with the SPR test results.

Overall, the manuscript requires substantial revisions, improvement in English language and sentence structures.

Response:The manuscript has been revised to meet the requirements of the magazine as much as possible.

Reviewer 2 Report

This paper is interesting enough and has described about "virtual screening and molecular dynamics simulation study of 2 influenza polymerase PB2 inhibitors". However, in order to be considered for publication in Molecules, there are some points should be consider by authors:

1. Abstract line 22-23, Kd is written as 0.21-6.77 M, while on line 158-159, it is 0.081-1.53 ​​M, as well as data in Tabel 1. It is likely just copy paste from published paper in Molecules, 2020, 25(22).
Please be careful.

2. Receptor selection and preparation (line 84-85); the protein used with ID of 6EUV, while at line 414 it was written ID 5WOL. Pleas recheck it!

3. Verification of docking method (line 101-103) XP GScore -10.440. The XP GScore cutoff used is <-0.7 (lines 118 and 128) while the method (lines 438 and 441) is <-0.9. An explanation is needed regarding the consideration of using the cutoff value.

4. ADME property prediction (line 120): the parameters analyzed are only absorption and distribution. It is recommended that all ADME parameters be analyzed by comparison with the VX-787 compound based on the background, the development is currently in phase 3 testing, but there are obstacles to pharmacokinetic characteristics that are not good for further development (line 75-76). Based on Table 1 (line 172), Str1614 compounds with the best Kd (close to VX-787), and stated to have good binding affinity (These results indicated that our compounds have good binding affinities for the influenza virus PB2 CBD on lines 161-162). has ADMET Absorption Level value 2 (low), which means its pharmacokinetic characteristics are also not good.

5. Line 131-133: If there is no compound that meets the requirements in the cluster, but the XP GScores of compounds in this cluster are generally high, then compound is also selected. The lower the XP Score the better, it should be "generally low".

6. Line 334: PPb2 CBD, should be PB2 CBD.

7. Table 2 (line 342): Please check the table caption.
Asterisk is added for every increase of 0.5 in the interaction fraction value (line 340). In Table 2 there is an asterisk 8 data, which means the interaction fraction becomes 4?
Line 342-362 description does not indicate the compound in question.
In order to state the amino acid residues that are important in the interaction, there should be a description of the references or previous studies that state these amino acids are important for activity (lines 365-408). It is necessary to add how it relates to activity, if the amino acid residues of the protein interacting with the active compound and VX-78 as a comparison are different. 

8. Before Figure 3 and 4, it is better to present one figure consist of the graph of RMSD plot (backbone of receptor, ligand), and RMSF plot (residue of receptor).

Author Response

Dear Sir or Madam: 

Thanks a lot for your attention to our manuscript and we appreciate your professional and detailed criticisms and suggestions. Thank you very much for your helpful suggestion. The manuscript has been extensively revised according to your comments. For the changes made according to reviewer suggestions, we have marked them in yellow in the text. The rewritten contents have been marked in blue. The corresponding explanations of each point which is raised in your comments as follows:

Comments:  1. Abstract line 22-23, Kd is written as 0.21-6.77 M, while on line 158-159, it is 0.081-1.53 ​​M, as well as data in Tabel 1. It is likely just copy paste from published paper in Molecules, 2020, 25(22).
Please be careful.

Response:Thank you for your careful correction. We have corrected the data, and the revised part is marked in yellow.

Comments: 2. Receptor selection and preparation (line 84-85); the protein used with ID of 6EUV, while at line 414 it was written ID 5WOL. Pleas recheck it!

Response:Thank you for your careful review, the ID of the protein we used should be 6EUV, and the error data in line 414 has been corrected.  

  1. Verification of docking method (line 101-103) XP GScore -10.440. The XP GScore cutoff used is <-0.7 (lines 118 and 128) while the method (lines 438 and 441) is <-0.9. An explanation is needed regarding the consideration of using the cutoff value.

Response: When verifying the docking method, the XP GScore of the VX-787 was -10.440 (lines 138 of the revised manuscript). When selecting compounds, the XP GScore value cutoff was set to -7.0. The XP GScore cutoff value in lines 138, and 506 of the revised manuscript should be <-7.0 instead of <-9.0, and the error value has been corrected and marked in yellow.

  1. ADME property prediction (line 120): the parameters analyzed are only absorption and distribution. It is recommended that all ADME parameters be analyzed by comparison with the VX-787 compound based on the background, the development is currently in phase 3 testing, but there are obstacles to pharmacokinetic characteristics that are not good for further development (line 75-76). Based on Table 1 (line 172,) Str1614 compounds with the best Kd (close to VX-787), and stated to have good binding affinity (These results indicated that our compounds have good binding affinities for the influenza virus PB2 CBD on lines 161-162). has ADMET Absorption Level value 2 (low), which means its pharmacokinetic characteristics are also not good.

Response: Discovery Studio software can predict six ADMET predictors of compounds, including aqueous solubility, blood-brain barrier penetration, CYP2D6 binding, liver toxicity, intestinal absorption and plasma protein binding. The main purpose of this article is to find lead compounds through a combination of in silico structure-based virtual screening approaches and biological evaluations. Therefore, when selecting compounds, we only investigated two predictors: aqueous solubility and intestinal absorption (mainly related to membrane permeability), which are related to the in vitro activity of the compound. The other four predictors, including blood-brain barrier penetration, CYP2D6 binding, liver toxicity, and plasma protein binding, have little effect on the in vitro activity of the compound. At the same time, we also did not conduct relevant tests on these properties. So, we did not analyze these predictors when selecting compounds. And we actually screened very few compounds (only more than 8000), if we consider more predictors, we may not find active compounds.

Indeed, as you said, the compound Str1614 has a good binding affinity to the influenza PB2 CBD, but its cellular level antiviral activity is not high. This may be related to its low membrane permeability or/and low solubility. However, since there is no experimental data to support this prediction, this article does not discuss it.

  1. Line 131-133: If there is no compound that meets the requirements in the cluster, but the XP GScores of compounds in this cluster are generally high, then compound is also selected. The lower the XP Score the better, it should be "generally low".

Response:Thanks for your careful correction, corresponding correction has been made and marked in yellow. These contents have been adjusted to section 4.1.5 "In silico ADME analysis and compound selection" (line 517 of the revised manuscript).

  1. Line 334: PPb2 CBD, should be PB2 CBD.

Response:Thanks for your careful correction, corresponding correction has been made and marked in yellow. (line 400 of the revised manuscript). 

  1. Table 2 (line 342): Please check the table caption.

Response: The table caption has been modified, "the active" has been changed to "the active compounds"

Asterisk is added for every increase of 0.5 in the interaction fraction value (line 340). In Table 2 there is an asterisk 8 data, which means the interaction fraction becomes 4?

Response:Yes, please see Figure 5F in the revised manuscript.

Line 342-362 description does not indicate the compound in question.

Response:The compounds discussed in lines 342-362 (line 413 -431 of the revised manuscript) refer to the six active compounds we found. In order to clarify this point, we changed the text " It can be seen from Table 2 that although the binding modes of the six active compounds differ” " in the original text to " It can be seen from Table 2 that although the binding modes of the six active compounds we found differ”

In order to state the amino acid residues that are important in the interaction, there should be a description of the references or previous studies that state these amino acids are important for activity (lines 365-408). It is necessary to add how it relates to activity, if the amino acid residues of the protein interacting with the active compound and VX-787 as a comparison are different. 

Response:

Paragraph 3.2 of (Analysis of PB2-ligand binding model based on dynamic simulation results,lines 318-396 the revised manuscript) have discusses the binding mode of VX-787 to PB2 in detail in conjunction with the literature(lines 330-339 the revised manuscript), and clearly pointed out that amino acid residues, such as Lys376, Glu361, His357, Phe323, Phe363, Phe404, His357, Asn510, Arg355, Val511, are essential for maintaining the binding of VX-787 and PB2(thus are important for activity). At the same time, in this paragraph (lines340-387 of the revised manuscript), we also discussed and analyzed the binding modes of the active compounds we found in more detail, and clarified the amino acid residues that play a key role in the binding of each compound to the PB2 protein. We found that the unique binding modes of our active compounds are distinct from that of VX-787, and the same amino residue may play different roles when interacting with different ligands. Therefore, it is difficult for us to summarize the contribution of a certain amino acid residue to the activities of all ligands. Therefore, in paragraph 3.3 (key amino acid residues of PB2 CBD analysis based on dynamic simulation results, lines 397-478 of the revised manuscript), we only discussed those amino acid residues that play a key role in maintaining the binding of the protein to the ligand, and did not discuss the mode of action of these amino acid residues.

  1. Before Figure 3 and 4, it is better to present one figure consist of the graph of RMSD plot (backbone of receptor, ligand), and RMSF plot (residue of receptor).

Response:The RMSD of PB2 protein Cα atoms and Lig fit Prot of ligands in the MD simulations have been provided in Supplementary Figure S4. The root-mean-square fluctuation (RMSF) of amino acid residue in the MD simulations are presented in Supplementary Figure S5 (lines 211- 222 of the revised manuscript). Due to the length of the article, they are not provided in the text.

Round 2

Reviewer 1 Report

My comment: It appears the entire section 3.3 did not cite any literature to support the results and it appears so in other sections in the manuscript.

Response of authors: The main content of section 3.3 is to report the work of finding the dominate amino acid residues that play a key role in the binding of ligand and protein by analyzing the results of molecular dynamics simulation. This work requires little background knowledge, and there are no reports of similar work on PB2 inhibitors, so we did not cite any literature in this part of this article.

My comment:

I still believe that the authors need literature from similar works to support the section 3.3. Maybe the there is limited work on the PB2 inhibitors, but articles can be obtained to support some of the inferences made. The section 3.3 must be improved.

Author Response

Dear Sir or Madam:

Thanks a lot for your attention to our manuscript and we appreciate your professional and  criticisms and suggestions. Thank you very much for your helpful suggestion. We are pleased to follow the referee’s suggestions and criticisms. The manuscript has also been extensively revised according to the comments. For the changes made according to reviewer suggestions, we have marked them in yellow in the text. 

.

Kind regards,

Xingzhou Li

2021-11-10
